# Interface-driven energy-independent charge extraction in GaN photocatalysts

Yuying Gao [1,2,5] ✉, Yuxin Xie[2,3,5], Christian Höhn[1], Markus Wollgarten [1], Holger Kropf[1], Fengtao Fan [2], Can Li [2], Roel van de Krol [1,4] & Dennis Friedrich [1] ✉

Ultrafast charge transfer dynamics are key to photocatalytic efficiency, governing energy relaxation and surface reactivity. However, the temporal evolution of carrier energy landscapes following photoexcitation, particularly at complex metal/semiconductor interfaces, remains poorly understood. Here, we present a surface- and energy-resolved investigation of ultrafast electron dynamics across bare and Pt-modified gallium nitride (GaN) surfaces using time-resolved two-photon photoemission spectroscopy. We show that photogenerated electrons rapidly thermalize to the conduction band minimum and undergo sub-picosecond trapping in nitrogen-vacancy-related surface states. Surface modification with Pt suppresses these trapping channels and introduces an energy-independent ultrafast electron transfer pathway (~50 fs) from GaN into Pt. By disentangling interfacial charge transfer from intrinsic relaxation mechanisms through tailored pump-probe configurations, we demonstrate that Pt facilitates picosecond-scale electron transport from the bulk to the surface by photoinduced dynamic band flattening. Modulating these ultrafast dynamics through interfacial engineering significantly enhances charge separation and photoelectrochemical performance. This study deepens the understanding of interface-dependent relaxation and transfer processes of photocarriers and provides valuable guidance for rational design of advanced photocatalytic systems.

Solar-to-chemical energy conversion offers a promising pathway toward sustainable fuel production and long-term energy security[1–3]. Central to advancing semiconductor-based artificial photosynthesis is the ability to understand and control the dynamics of photoexcited charge carriers[4,5]. These carriers undergo rapid energy relaxation, trapping, and transport on ultrafast timescales before reaching reactive sites, where they drive chemical transformations on slower millisecond timescales[6]. Importantly, the energy relaxation dynamics of these nonequilibrium carriers strongly influence their chemical reactivity, as only carriers exceeding specific energy thresholds can

overcome activation barriers associated with key redox reactions[2,7]. For example, photogenerated electrons with energies above 0 eV vs. RHE are required for hydrogen evolution, while photogenerated holes with energies exceeding 1.23 eV vs. RHE at pH 0 are necessary for water oxidation in water splitting[1]. Although significant progress has been made in understanding photo-induced charge transfer dynamics through time-resolved spectroscopy[6,8–11], most studies have focused on bulk carrier behaviors or averaged interfacial processes, whereas the femtosecond energy relaxation pathways at metal/semiconductor interfaces remain poorly understood. In particular, it is not well

[1]Institute for Solar Fuels, Helmholtz-Zentrum Berlin für Materialien und Energie GmbH, Berlin, Germany. [2]State Key Laboratory of Catalysis, Dalian National Laboratory for Clean Energy, Dalian Institute of Chemical Physics, Chinese Academy of Sciences, Dalian, China. [3]School of Chemistry and Materials Science, University of Science and Technology of China, Hefei, China. [4]Institut für Chemie, Technische Universität Berlin, Berlin, Germany. [5]These authors contributed equally: Yuying Gao, Yuxin Xie. ✉e-mail: yuying.gao@helmholtz-berlin.de; friedrich@helmholtz-berlin.de

established how surface modifications alter the initial energy distribution, defect trapping, and interfacial injection of hot carriers immediately after photoexcitation. This knowledge gap is critical because only carriers that retain sufficient excess energy on ultrafast timescales can efficiently participate in catalytic reactions. Addressing this gap is essential for guiding the rational design of photocatalytic systems capable of preserving hot carrier energy long enough to couple effectively with surface reactions.

Gallium nitride (GaN) has emerged as a compelling platform for solar fuel production, as demonstrated by previous studies on photoelectrochemical (PEC) water splitting[12–14], $CO_2$ reduction[15,16], and ammonia decomposition[17]. The separation and transport of photogenerated carriers in GaN are intrinsically influenced by factors such as the bulk electronic structure, crystal facet orientation, morphology, and surface states[18–20]. Surface modification of GaN with metal cocatalysts such as platinum (Pt)[14] or cobalt (Co)[21] has been shown to dramatically enhance PEC performance. These hybrid catalysts benefit from engineered metal/semiconductor interfaces that optimize interactions between catalytic sites and reaction intermediates, reduce kinetic barriers, and improve charge separation. Despite these advances, the mechanistic details of how such metal/semiconductor interfaces reshape the ultrafast carrier energy landscape remain elusive. Specifically, the formation of a metal/semiconductor junction introduces interfacial states[14,22] that modulate the built-in electric field and band alignment[23,24], thereby influencing the carrier energy landscape and transport behavior. Yet these effects have rarely been studied with sufficient energy and time resolution to capture energy-resolved ultrafast charge dynamics.

Herein, we employ time-resolved two-photon photoemission (tr-2PPE) spectroscopy to directly probe the energy-resolved femtosecond dynamics of hot electrons at bare and Pt-modified $n$-GaN surfaces. Our measurements reveal ultrafast thermalization of hot electrons to the conduction band minimum (CBM), followed by rapid trapping into deep-defect states near the Fermi level ($E_F$) on sub-picosecond timescales, leading to long-lived states. Upon Pt modification, we observe a significant suppression of these defect-mediated processes, alongside the emergence of an ultrafast charge transfer pathway characterized by energy-independent electron injection into Pt within ~50 fs. By combining one-color and two-color excitation schemes, we disentangle the competing contributions of surface trapping, interfacial charge transfer, and bulk-to-surface transport under tunable excitation conditions. We find that Pt decoration enhances bulk-to-surface electron transport in the conduction band, facilitated by ultrafast interfacial electron injection and dynamic flattening of the surface band bending. The dynamic interplay of these effects promotes charge separation, significantly boosting the photoelectrochemical hydrogen evolution efficiency. Our results provide direct mechanistic insight into how interface engineering preserves hot carrier energy and enhances bulk-to-surface transport, offering a framework for rational design of cocatalysts and semiconductor interfaces to improve photocatalytic efficiency.

## Results and discussion
### tr-2PPE characterization of GaN photocatalyst

Figure 1a illustrates the experimental setup for time-resolved two-photon photoemission (tr-2PPE) spectroscopy used to investigate ultrafast electron dynamics in both temporal and energy domains. In this technique, a pump pulse excites electrons to generate a population of excited states, while a time-delayed probe pulse photoemits these electrons above the vacuum level. The resulting photoelectrons are energy-analyzed using a time-of-flight (TOF) detector.

Our tr-2PPE setup incorporates two ultrafast ultraviolet (UV, 4.49 eV) laser pulses and one visible (Vis, 2.58 eV) laser pulse, with their relative delay precisely controlled by two variable delay stages. The distinct photon energies enable two complementary pump-probe configurations to achieve state-resolved photoexcitation. In the one-color (UVUV pump-probe) scheme, photoemission arises from band-to-band transitions in bulk GaN (bandgap: 3.4 eV) and from transitions involving (occupied) surface states. In contrast, the two-color (VisUV pump-probe) configuration enables selective probing of electron populations associated with surface defect states distributed within the bandgap (see "Methods" for further details). These configurations provide rich spectral access to nonequilibrium carrier distributions under tunable excitation conditions, as illustrated below.

The inset in Fig. 1a depicts the energy band structure of the n-GaN surface with low Mg doping concentration, which was cleaned through three cycles of $Ar^+$ sputtering followed by annealing at 850 °C in ultrahigh vacuum. This process resulted in a well-ordered (1 × 1) surface reconstruction, as confirmed by low-energy electron diffraction (Supplementary Fig. 1), and produced an atomically flat surface (Supplementary Fig. 2). Note that although Mg is an acceptor-type dopant, the GaN remains n-type due to the high activation energy of Mg acceptors and the low doping concentration used here. Ultraviolet photoelectron spectroscopy (UPS) measurements indicate that the valence band maximum (VBM) is located 2.51 eV below the Fermi level ($E_F$) (Supplementary Fig. 3), consistent with previous reports on Mg-doped n-type GaN surfaces[25]. Raman spectroscopy identifies a high-frequency longitudinal optical phonon-plasmon coupled (LOPC) mode at 748 $cm^{-1}$ (Supplementary Fig. 4), indicating n-type conductivity as the concentration of Mg acceptors is not high enough to induce effective p-type behavior[26]. This is further corroborated by the observation of a positive surface photovoltage response under bandgap excitation (Supplementary Fig. 5), which is characteristic of n-type surfaces. In addition, UPS spectra reveal a continuous filled electronic band arising from near-surface states extending from the VBM up to $E_F$ (Supplementary Fig. 3b), leading to an upward band bending.

We first investigated the photoelectron dynamics under one-color UVUV excitation conditions. In this configuration, both the pump and probe pulses possess the same photon energy, with the pump pulse set to half the power of the probe at positive time delays. The fluence of the UV pump pulse was carefully controlled to a low value (~4 μJ/$cm^2$), corresponding to an excitation density of $2 \times 10^{17}$ $cm^3$, thereby avoiding complications from multiple photon excitation and Auger recombination[27]. Figure 1b presents a pseudo-color plot of the tr-2PPE spectra of bare GaN after subtracting the background signal measured at −5 ps (the raw data is shown in Supplementary Fig. 6). The energy scale is referenced to $E_F$ to facilitate visualization of the excited-state energy distribution. The spectra reveal a symmetric and short-lived photoemission feature extending up to 3.5 eV above $E_F$, along with a lower-energy component around $E_F$ that exhibits a significantly longer lifetime. At a delay time of 0 ps (Fig. 1c), a pronounced photoemission peak is observed near 0.06 eV above $E_F$, accompanied by a weaker but discernible broad spectral feature spanning the higher energy range (0.5 eV ≤ $E - E_F$ ≤ 3.5 eV, see inset of Fig. 1c). Given the wide bandgap of GaN (3.4 eV) and the measured surface work function of 4.19 eV (Supplementary Fig. 3a), this broad continuum is attributed to two-photon photoemission from occupied surface states lying below $E_F$ (Supplementary Fig. 7), in agreement with the UPS results. The pronounced low-energy peak is assigned to defect-related surface states near $E_F$, populated via photoexcitation from the GaN valence band. These defect states are likely associated with interstitial magnesium ($Mg_i$), which forms readily at low Mg doping levels due to incomplete incorporation on Ga sites[28]. These $Mg_i$ species can act as donor-like traps, introducing occupied electronic states below $E_F$. Notably, as the delay time increases, this peak shifts by 85 meV towards lower energies, falling below $E_F$, which corresponds to the equilibrium distribution of occupied states. Moreover, the peak intensity initially decreases within the first 0.05 ps, followed by a gradual rise at longer delay times. The subsequent discussion will elaborate on this behavior,

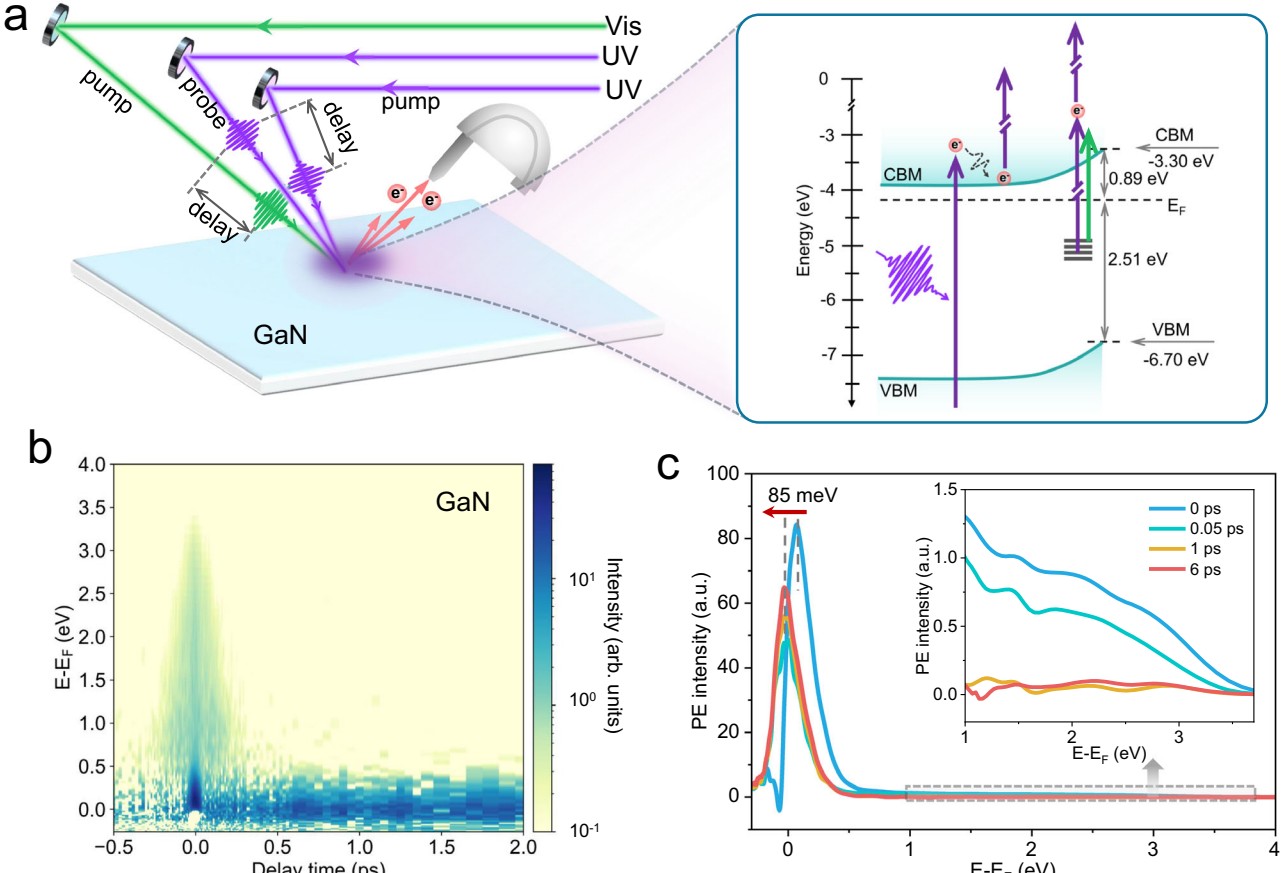

**Fig. 1 | Time-resolved 2PPE measurements for GaN. a** Schematic illustration of the tr-2PPE experimental setup, showing both the one-color (UV-UV) and two-color (Vis-UV) configurations. Two delay stages were employed to control the delay times between the UV-UV and Vis-UV pulses. The right panel depicts the schematic energy band diagrams of bare GaN and the corresponding photoemission processes. CBM conduction band minimum, VBM valence band maximum, $E_F$ Fermi level. **b** Representative two-dimensional pseudo-color 2PPE spectrum of GaN obtained under one-color excitation with a photon energy of 4.49 eV (276 nm), after background subtraction using the signal recorded at −5 ps. **c** Photoemission (PE) intensity as a function of energy relative to $E_F$ for GaN at various delay times. The inset shows an enlarged view in the energy range of 1.0–3.8 eV.

attributing the peak shift to the photoexcitation dynamics associated with surface photovoltage (SPV).

**Ultrafast dynamics of defect-assisted charge transfer and SPV**
To resolve the temporal evolution of excited states, Fig. 2a presents representative time-resolved 2PPE traces at four characteristic energies: 2.5 eV, 1.0 eV, 0.5 eV, and −0.2 eV, corresponding to photoemission from the conduction band (CB), conduction band minimum (CBM), defect band (DB), and below $E_F$ induced by peak shift (PF), respectively. Each trace represents the integrated photoemission intensity within ±0.2 eV around the specified energy. The transient responses exhibit distinctly different dynamics across these energies. At 2.5 eV, excited electrons in the CB undergo rapid thermalization, characterized by a decay constant of 0.07 ps (see "Methods" for fitting details), leading to fast energy redistribution toward lower-lying CB states. At the CBM, the photoemission profile exhibits a sharp peak at time zero due to pump–probe temporal overlap, followed by a slower relaxation with a time constant of 0.12 ps, consistent with previously reported hot electron cooling dynamics in ZnO[29] and InP[30]. This sub-picosecond decay is associated with the thermalization of hot electrons from higher CB states and ultrafast trapping by defect states within the DB.

In contrast, the transient feature at 0.5 eV is attributed to mid-gap DB states associated with nitrogen vacancies[28], as further supported by electron paramagnetic resonance (EPR) spectra revealing a pronounced deep donors signal related to nitrogen vacancies

(Supplementary Fig. 8). These defect-related features in GaN (see Supplementary Fig. 9a for detailed defect distributions) differ significantly from the characteristic gradual population typically observed during defect-state trapping, as observed in $Cu_2O$[31]. Instead, we observe a pronounced and sharp emission peak with a temporal width comparable to the autocorrelation (AC) trace of pump-probe pulses, followed by a slower decay with a characteristic lifetime of 0.38 ps. These observations indicate that photoemission from the DB states proceeds via a sequential one-photon-driven excitation pathway involving a real populated intermediate state, rather than a virtual intermediate state with negligible lifetime[32]. This provides direct evidence for ultrafast electron trapping into deeper-lying defect levels near $E_F$, as indicated by the time-dependent increase in photoemission intensity at this energy (Fig. 1b). Moreover, the observed fluence independence of this slower decay process further supports the assignment to an ultrafast defect capture mechanism (Supplementary Fig. 9b–d)[33].

Interestingly, the dynamics at −0.2 eV relative to $E_F$ exhibit an initial intensity dip (Fig. 2a), followed by a rapid rise (~33 fs) comparable to the pump-probe AC width (~40 fs), and then a gradual increase toward saturation over 2.28 ps. This behavior suggests the emergence of electronic states below $E_F$ induced by photoexcitation. Given the low exciton binding energy in GaN (~23 meV)[34], exciton formation is unlikely under ambient conditions. Moreover, the fluence-dependent measurements reveal a more pronounced ultrafast energy shift of this feature at higher excitation densities (Fig. 2b), further ruling out an

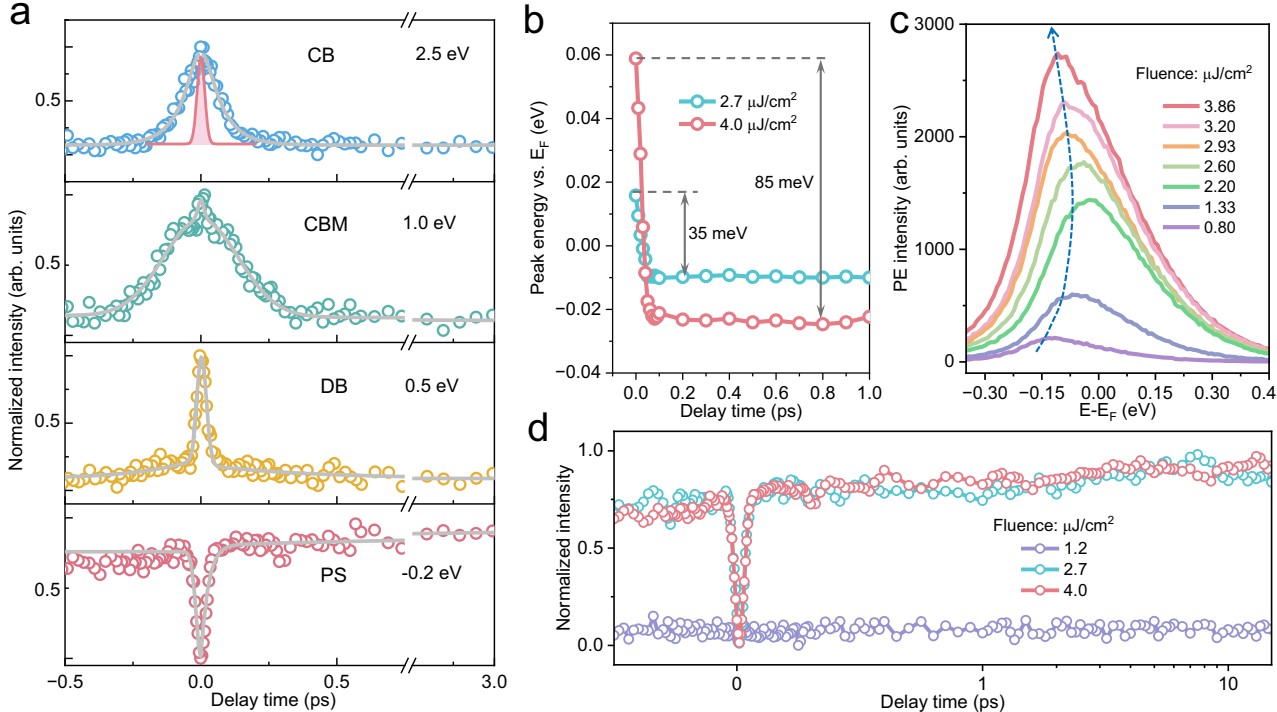

**Fig. 2 | Energy-resolved ultrafast charge relaxation dynamics. a** Normalized dynamical trace (open circles) of energy-resolved photoemission intensity (PE) at four representative excited electron energies relative to $E_F$. The gray lines represent fits to the kinetic model used to extract time constants, while the shaded red line denotes the autocorrelation of the pump-probe pulses. CB: conduction band; CBM: conduction band minimum; DB: defect band; PS: peak shift. **b** Peak energy of the 2PPE spectra as a function of delay time under different UV laser fluences. **c** 2PPE spectra recorded at zero delay time under varying UV fluences. The dashed arrow indicates the emission peak shift. **d** Normalized dynamical traces at an energy of −0.2 eV relative to $E_F$ obtained at three different fluences. The PE dip at time zero, attributed to the PS, disappears at low fluences as the SPV response becomes dominated by negative signals from electron trapping in surface defect states.

excitonic origin[29]. Additionally, the absence of asymmetric peak shapes or phonon-spaced replica bands (Fig. 1c) excludes the possibility of polaron formation[35,36]. We thus attribute the signal below $E_F$ to a photoinduced shift of pre-existing trap states near $E_F$, driven by the SPV effect.

We performed 2PPE spectroscopy focusing on the emission peak energy under varying pump fluences. With increasing pump fluence, the peak energy initially increases and subsequently decreases (Fig. 2c), indicating the coexistence of two competing SPV mechanisms. The initial positive shift (-SPV) arises from photogenerated electron trapping in surface defect states, while the subsequent negative shift (+SPV) is attributed to interband excitation in n-type GaN, as corroborated by SPV spectroscopy (Supplementary Fig. 5a–c). This interpretation is further supported by simplified simulations (Supplementary Fig. 5d, e), which reproduce the experimentally observed band flattening and show that bandgap illumination reduces the upward band bending at the GaN surface by approximately 0.1 eV. The transient signal observed below $E_F$ arises from a transient reduction in upward surface band bending upon bandgap excitation, leading to a rigid downward shift of the energy band edge, consistent with dynamic SPV effects. This feature vanishes at lower excitation fluence (Fig. 2d), as under these conditions the SPV response is predominantly governed by defect-mediated negative signals, further supporting its assignment to photoinduced SPV.

## Modulation of charge transfer dynamics through surface modification

The vectorial superposition of the two SPV components described above leads to a net reduction in the surface photovoltage. This reduces the extent of band bending near the surface, which in turn narrows the space charge region and diminishes the selectivity for extracting photogenerated carriers, ultimately limiting the efficiency of solar energy conversion[37]. Surface functionalization with metal nanoparticles such as Pt[14,24] or Au[38], presents a promising strategy to overcome this limitation. By forming metal/semiconductor contacts, these nanoparticles induce additional band bending at the interface, thereby expanding the width of the space charge region. This modulation of the internal electric field enhances charge separation by promoting directional carrier extraction and reducing recombination losses. Motivated by this approach, we fabricated Pt-decorated GaN samples (denoted as Pt/GaN) to explore how surface modification modulates the hot carrier relaxation pathways and charge transfer dynamics.

Atomic force microscopy (AFM, Fig. 3a) reveals that Pt forms uniformly distributed nanoislands on the GaN surface, with an average height of ~0.65 nm (Fig. 3b), which is smaller than the photoelectron escape depth (~4 nm) at the excitation energy (4.49 eV)[39]. Statistical analysis of AFM images (Supplementary Fig. 10a) reveals that Pt predominantly forms discrete nano-islands with an average height size of ~0.73 nm and a surface coverage of ~56%. The homogeneous distribution across the surface indicates that charge transfer occurs primarily through localized metal/semiconductor junctions rather than continuous metallic channels. Transmission electron microscopy (TEM, Supplementary Fig. 10b, c) images show that the Pt nanoparticles are uniformly anchored on the GaN substrate with intimate interfacial contact. Moreover, scanning transmission electron microscopy (STEM, Supplementary Fig. 10d) image reveals a sharply defined, microscopically flat, and structurally uniform Pt/GaN interface. The partial Pt coverage allows us to detect photoemission signals from Pt, GaN, and their interface. Surface modification with Pt results in an increased work function of GaN from 4.19 eV to 4.43 eV (Supplementary Fig. 10c, d). High-resolution X-ray photoelectron spectroscopy

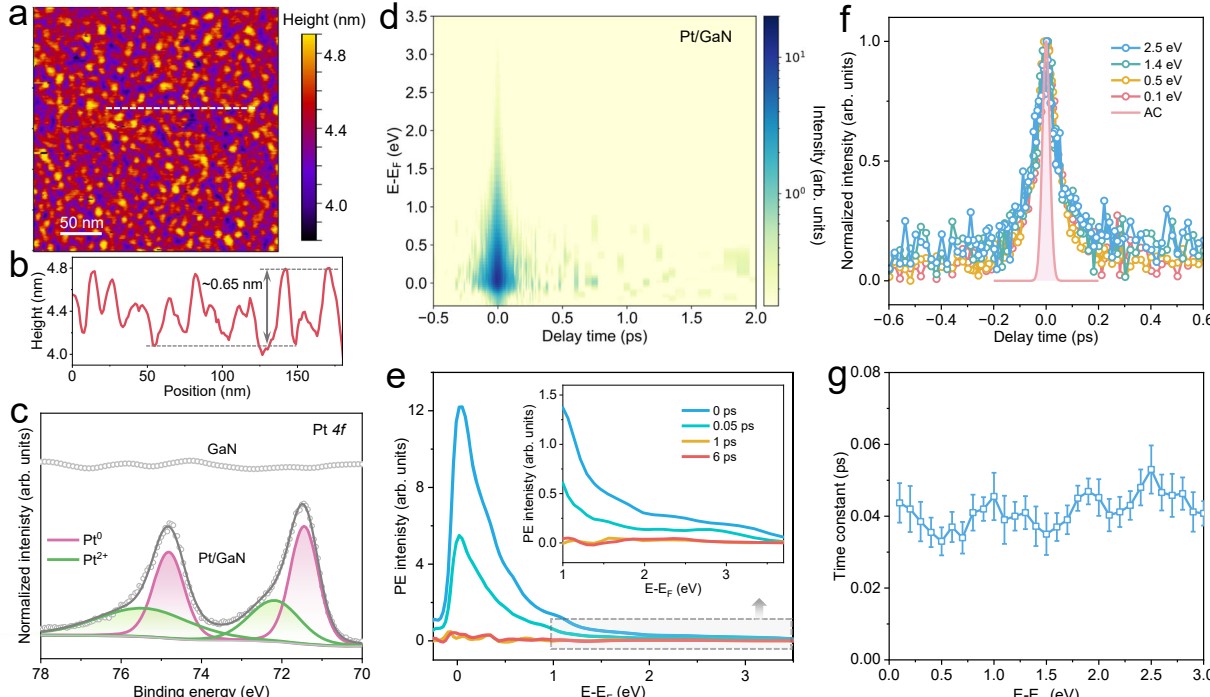

**Fig. 3 | Energy-independent interfacial electron transfer. a** Atomic force microscopy (AFM) image of GaN after surface Pt modification. **b** Height profile along the dashed white line in a, showing Pt nanoislands with an average height of approximately 0.65 nm. **c** High-resolution Pt $4f$ X-ray photoelectron spectroscopy (XPS) of GaN and Pt/GaN samples. **d** Representative two-dimensional pseudo-color 2PPE spectrum of Pt/GaN acquired under one-color excitation at a photon energy of 4.49 eV (276 nm) after background subtraction using the signal recorded at −5 ps. **e** 2PPE spectra of Pt/GaN recorded at various pump-probe delay times. The inset displays an enlarged view within the energy range of 1.0–3.8 eV. **f** Temporal evolution of the normalized photoemission intensity for excited-state carriers at selected energies. The shaded red curve represents the autocorrelation trace of the pump and probe pulses. **g** Extracted time constants as a function of excited-state energy, revealing an energy-independent interfacial electron transfer process. Error bars represent the standard deviation of the fits.

(HR-XPS) of Pt $4f$ (Fig. 3c) indicates the predominance of metallic Pt, along with a minor contribution from $Pt^{2+}$ species, likely arising from Pt–N bonding at the Pt/GaN interface[14]. Moreover, the characteristic binding energies of N $1s$ and Ga $2p_{3/2}$ in Pt/GaN exhibit a positive shift relative to those in pristine GaN (Supplementary Fig. 11), indicative of interfacial charge redistribution and the formation of strong electronic coupling at the metal–semiconductor interface.

Figure 3d presents the background-subtracted pseudocolour 2PPE spectrum of Pt/GaN acquired under one-color configuration (4.49 eV, see raw data in Supplementary Fig. 12). At time zero, a symmetric transient photoemission appears near the Fermi level, extending up to 3.5 eV, and decays within 0.3 ps. This signal arises from two-photon absorption (Supplementary Fig. 13). Notably, unlike bare GaN, the emission associated with long-lived defect trapping is fully suppressed, indicating that Pt decoration effectively passivates surface trap states. Nevertheless, the initial spectral profile of Pt/GaN remains similar to that of bare GaN (Fig. 3e), featuring a pronounced emission peak just above $E_F$ (-0.03 eV) and a weak high-energy tail ($E − E_F > 1$ eV). This suggests that Pt modification does not substantially alter the distribution of occupied surface states and optical transition channels. As the delay increases, no spectral shift of the emission peak is observed, likely due to suppression of electron emission below $E_F$ by vacuum states[30], as evidenced by the asymmetry of the peak.

Kinetic analysis of energy-sliced 2PPE spectra (Fig. 3f) reveals identical decay dynamics across excitation energies. Clearly, the relaxation dynamics in Pt/GaN extend beyond the pump-probe autocorrelation window (shaded red curve), confirming that the observed kinetics are not limited by the instrument's temporal resolution. Fitting the time-dependent signals (see "Methods" for details) yields a time constant of $41 \pm 13$ fs (Fig. 3g). Notably, the lifetimes of excited electrons typically exhibit strong energy dependence in both

semiconductors[29,30,40] and metals[41,42]. The absence of such dependence here provides compelling evidence for a uniform ultrafast energy-independent electron transfer from GaN to Pt, occurring within 41 fs across both DB and CB states.

## Photoemission signatures of enhanced bulk-to-surface electron transport

We next performed tr-2PPE measurements using a two-color configuration with a Vis pulse (2.58 eV, 480 nm) and a UV pulse (4.49 eV, 276 nm) at a 25:1 intensity ratio. In this scheme, positive delay times correspond to the Vis pulse (pump) arriving prior to the UV pulse (probe), while negative delay times represent the UV pulse (pump) arriving after the Vis pulse (probe). To account for differing probe photon energies across the delay axis, all photoemission signals are referenced to kinetic energy, as illustrated in Fig. 4a, b for the background-subtracted spectra for GaN and Pt/GaN data (raw data as shown in Supplementary Figs. 14 and 15). The background signal arises primarily from single-photon UV photoemission near $E_F$ and two-photon processes induced by the Vis pulse (Supplementary Figs. 16 and 17).

At time zero, both samples exhibit similarly strong transient emission features with dynamics exceeding the pump-probe cross-correlation responses (Supplementary Fig. 18), indicating that the photoemission arises from real intermediate states. These peaks are attributed to initial excitation of VB states induced by both UV and Vis photons, as well as occupied DB states populated via UV excitation (Supplementary Fig. 19 and Table S1). Notably, the relaxation dynamics diverge significantly at longer delay times, particularly within the low kinetic energy region ($E_{kin} \leq 1.0$ eV, Supplementary Fig. 18), suggesting a marked influence of Pt surface functionalization on hot-carrier behavior. Analysis of the integrated photoemission intensity within the

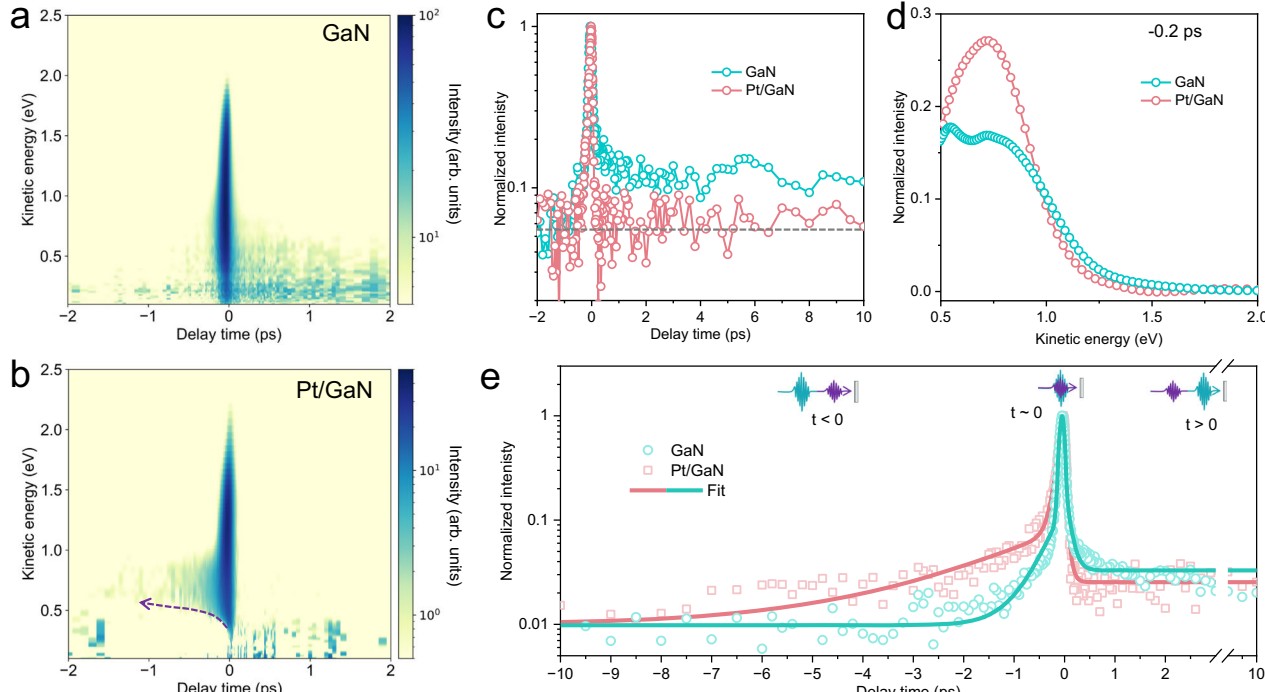

**Fig. 4 | Signature of bulk-to-surface charge migration. a, b** Representative two-dimensional pseudo-color 2PPE spectra of (**a**) GaN and (**b**) Pt/GaN acquired under two-color excitation with a UV photon energy of 4.49 eV (276 nm) and a visible (Vis) photon energy of 2.58 eV (480 nm), after background subtraction using the signal recorded at −10 ps. The dashed arrow in **b** indicates the shift of the photoemission edge toward higher energies with increasing negative delay times. **c** Temporal evolution of the normalized photoemission intensity for excited-state carriers at a kinetic energy of 0.5 ± 0.2 eV. The gray line denotes the background level. **d** Normalized 2PPE spectra of GaN and Pt/GaN recorded at a delay time of −0.2 ps. **e** Temporal traces of the normalized photoemission intensity at a kinetic energy of 0.90 ± 0.20 eV for GaN and 0.65 ± 0.20 eV for Pt/GaN. The solid lines represent fits to the kinetic model to extract the characteristic time constants.

0.5 ± 0.2 eV energy window (Fig. 4c) reveals a significant suppression of defect-mediated trapping processes in Pt/GaN at positive delay times. This observation, consistent with the one-color measurements, demonstrates that Pt decoration modifies the energy dissipation landscape of photoexcited electrons, efficiently redirecting them away from defect-assisted trapping towards faster interfacial charge transfer.

Interestingly, the slow carrier dynamics observed within the 0.5–1.0 eV energy window at negative delay times (Fig. 4a, b and Supplementary Fig. 18), particularly for Pt/GaN, deviate from typical behaviors in semiconductor systems[29–31,43] under two-color excitation, where UV excitation typically produces a sharp spectral edge at negative delays due to the its higher photon energy and the ultrafast decay of photoexcited carriers. The persistence of photoemission signals beyond this expected decay window suggests the presence of a previously unrecognized charge relaxation channel that extends charge lifetimes beyond conventional expectations.

To elucidate the origin of this unexpected behavior, we compared photoemission spectra at −0.2 ps for both GaN and Pt/GaN, with the signal intensities normalized to the corresponding UV fluence (Fig. 4d). This normalization accounts for the variations of pulse fluence on sample surfaces across measurements and enables a direct assessment of the fraction of long-lived carriers. Both samples exhibit similar spectral shapes, indicating comparable energy distributions, but the signal intensity is significantly higher in Pt/GaN, implying a greater population of long-lived carriers. These observations indicate that the delayed signal predominantly originates from GaN rather than Pt, in line with the energy-independent ultrafast electron transfer from GaN to Pt within 41 fs demonstrated above. Since surface defect trapping is suppressed in Pt/GaN, the presence of this signal at −0.2 ps in both samples suggests that it is unlikely to originate from surface-trapped

electrons in GaN. We therefore attribute this long-lived component to carrier transport from the GaN bulk to the surface. Given the intrinsic upward band bending of n-type GaN, such transport likely occurs in the diffusion-limited regime, providing a direct photoemission signature of bulk-to-surface charge migration.

These results indicate that Pt decoration on the GaN surface enhances the electron population involved in bulk-to-surface transport by approximately 50% (from 18% to 27%). This suggests that Pt not only provides an efficient pathway for interfacial charge transfer but also facilitates carrier extraction from the bulk. This enhancement arises from the synergistic interaction between bulk charge transport and ultrafast interfacial charge transfer: the rapid extraction of surface electrons by Pt alleviates electron trapping and accumulation at the GaN surface, thereby facilitating continuous charge diffusion from the bulk. This process is driven by a dynamic screening effect[44] and the photo-induced SPV, which transiently reduces the upward band bending upon femtosecond above-bandgap excitation[43]. This effect is evidenced by the progressive upward shift (-0.18 eV, dashed arrow in Fig. 4b) of the low-energy emission edge in Pt/GaN with increasing negative delay time. Notably, these observations are difficult to discern under a one-color configuration (Fig. 3d), but become evident under a two-color configuration due to the higher photon fluence and distinct emission characteristics associated with visible probing, highlighting the complementary nature of the two measurement schemes in capturing a more comprehensive picture of charge dynamics.

To elucidate the timescales governing bulk-to-surface carrier transport, we analyzed the temporal evolution of the normalized photoemission intensity within the kinetic energy of 0.9 eV±0.2 eV for GaN and 0.65 ± 0.20 eV for Pt/GaN, as shown in Fig. 4e. Global fitting of the time traces reveals that the charge dynamics in GaN can be well-described by a single-exponential decay convolved with a Gaussian

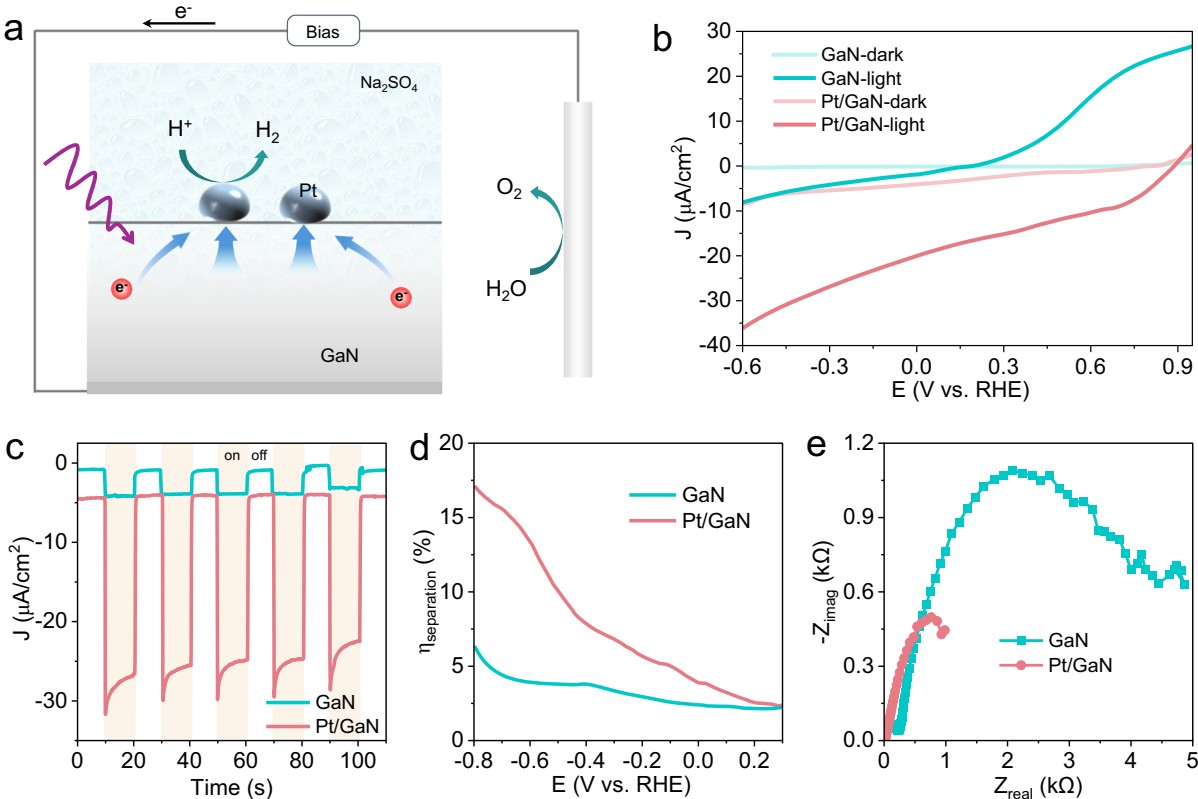

**Fig. 5 | PEC performance for HER. a** Schematic diagram of Pt/GaN photocathode for the PEC hydrogen revolution reaction. **b** Current density-voltage curves of the GaN and Pt/GaN samples measured under Xe-lamp illumination in 0.5 M $Na_2SO_4$. **c** Chronoamperometric I-t curves of the GaN and Pt/GaN samples measured at the applied potential of −0.4 V vs. RHE. **d** Calculated charge separation efficiencies of the GaN and Pt/GaN photoelectrodes under external bias. **e** PEIS curves of the GaN and Pt/GaN samples in 0.5 M $Na_2SO_4$.

instrument response for both positive and negative delay times. In contrast, a biexponential model is required to adequately capture the dynamics at negative delays for Pt/GaN. At positive delays, the extracted lifetime for GaN is $0.098 \pm 0.015$ ps, whereas a shorter lifetime of $0.059 \pm 0.015$ ps is observed for Pt/GaN, consistent with the characteristic timescale of the energy-independent ultrafast interfacial charge transfer process discussed earlier. At negative delays, the dynamics in GaN yield a surface electron relaxation time of $0.38 \pm 0.08$ ps, while Pt/GaN features a fast relaxation component of $0.065 \pm 0.022$ ps attributed to surface electron relaxation, and a slower component with a lifetime of $1.96 \pm 0.34$ ps associated with bulk-to-surface charge transport.

To gain deeper insights into the mechanism underlying the bulk-to-surface electron transport observed at negative delay times, we investigated the fluence dependence of carrier dynamics using tr-2PPE measurements, varying the UV pump fluences while maintaining a constant visible probe fluence (Supplementary Fig. 20a, b). Remarkably, Pt/GaN exhibits a pronounced increase in the transport time constants from the bulk to the surface in Pt/GaN with increasing UV photon fluence, deviating from the typical fluence-independent relaxation behavior associated with surface electrons in GaN. The findings from both one- and two-color experiments described above consistently confirm that surface defect-mediated trapping and relaxation are effectively suppressed in Pt/GaN. Although we cannot entirely rule out the involvement of bulk defect states, their contribution is expected to be limited, as supported by the fluence-independent defect-trapping dynamics observed in pristine GaN (Supplementary Fig. 21). We therefore attribute the observed fluence-dependent signal at zero delay time to an ultrafast positive SPV[45,46], which dynamically reduces the extent of surface band bending and

facilitates enhanced transport of photogenerated electrons towards the surface. At higher excitation fluences, a higher SPV leads to a transient flattening of the surface band bending, reducing the surface potential barrier and thereby promoting electron transport from deeper regions within the bulk to the surface. Interestingly, the energy distribution of surface-arriving electrons remains largely unchanged under different excitation fluences (Supplementary Fig. 20c), suggesting that this transport proceeds via a highly coherent, collective transport mechanism that is largely unaffected by scattering events or trap-mediated thermalization.

As summarized in Supplementary Fig. 20d, we present a schematic illustrating the ultrafast charge transfer and relaxation pathways in GaN and Pt/GaN under above-bandgap excitation. In pristine GaN, photoexcited electrons with excess energy rapidly thermalize to CBM within 0.38 ps and are subsequently trapped by deep surface states, leading to long-lived charge carriers (>10 ps). Upon surface modification with Pt, photoexcited electrons, regardless of their initial energy, are transferred to Pt in an ultrafast manner, altering both their energy distribution and thermalization behavior. This interfacial charge transfer mechanism efficiently prevents electron trapping at deep defect states, which are typically recognized recombination centers[6,47], thereby promoting efficient charge separation. Moreover, Pt enhances electron transport from the bulk to the surface, leading to an enhanced surface charge carrier density. These findings underscore the broader implications of Pt surface modification, which alters electron transport dynamics in a manner to enhance photocatalytic performance, where efficient charge separation and interfacial transport are essential[5,22,38].

Beyond the specific Pt/GaN system studied here, in which the metal and semiconductor exhibit distinct work functions, interfacial electron-transfer efficiency in GaN-based photocatalytic systems is

largely dictated by a combination of cocatalyst properties. Specifically, the semiconductor/cocatalyst band alignment, the chemical nature of interfacial bonding, and the ability to passivate surface states collectively dictate the kinetics and selectivity of interfacial electron transfer[48]. This broader perspective helps rationalize why a wide range of cocatalysts, including noble metals, transition metals, and metal oxides, can effectively facilitate carrier extraction on GaN surfaces.

## PEC hydrogen evolution performance

To understand the influence of interface-driven electron transport on steady-state photochemical reactions, we further explored a prototypical photoreduction process in a photoelectrochemical (PEC) configuration using a standard three-electrode setup under Xe lamp illumination. In this system, Pt/GaN heterostructures functioned as photocathodes, enabling the directional transfer of photogenerated electrons from GaN to the Pt surface, where the hydrogen evolution reaction (HER) occurs in a pH-neutral $Na_2SO_4$ electrolyte (Fig. 5a). While n-type GaN surfaces are typically associated with oxidation pathways, the selection of HER was also motivated by the thermodynamically favorable alignment between GaN's CBM (−3.3 eV) and the proton reduction potential (−4.5 eV). More importantly, focusing on an electron-mediated reduction reaction enables a direct correlation between the engineered interfacial electron-transfer channels and the resulting photocatalytic performance, establishing a design principle for interface engineering in photochemical systems.

Compared to pristine GaN (Fig. 5b), the Pt/GaN photocathode exhibited a significantly improved photocathodic current for HER, with the onset potential $V_{on}$ shifting from 0.16 V vs. RHE to 0.91 V vs. RHE. The $J$–$V$ curve of GaN displayed discernible anodic photocurrents at potentials above 0.2 V vs. RHE (Supplementary Fig. 22a), which can be attributed to an oxidation reaction occurring on the n-GaN surface. The presence of surface Pt allows the Pt/GaN photocathode to sustain photocathodic current for HER even at applied potentials exceeding 0.2 V. Photocathodic chronoamperometric (I-t) HER measurements at -0.4 V vs. RHE revealed that the photocurrent of Pt/GaN was 6.6 times higher than that of GaN photoelectrode (Fig. 5c). The Pt/GaN photocathode exhibits stable photocurrent for over 800 min under continuous illumination (Supplementary Fig. 22b), demonstrating good operational durability. Moreover, the Faradaic efficiency for $H_2$ evolution exceeds 97% at −0.4 V vs. RHE, confirming that nearly all photogenerated electrons are consumed in the HER process. Given the well-known catalytic activity of Pt toward the HER[14,49], we further assessed the charge separation efficiency ($\eta_{separation}$) to clarify the role of surface Pt modification in enhancing charge separation during the photochemical reaction in a $Na_2SO_4$ electrolyte containing $K_3[Fe(CN)_6]$ (see details in "Methods" and Supplementary Fig. 22c). Remarkably, the Pt/GaN photoelectrode achieved a substantially higher charge separation efficiency (-14%) compared to GaN (-4%) at the same applied potential of −0.6 V vs. RHE (Fig. 5d). Moreover, photoelectrochemical impedance spectroscopy (PEIS) was employed to analyze the charge transfer resistance at the electrode-electrolyte interface. The Nyquist plots (Fig. 5e) show that Pt/GaN exhibits markedly smaller semicircle diameters than pristine GaN, indicating more efficient interfacial charge transfer facilitated by Pt decoration. These results suggest that the synergistic improvement in bulk charge separation and interfacial charge transfer collectively contribute to improved charge separation efficiency and the observed increase in hydrogen evolution activity. The photogenerated charge separation and transport behaviors observed under UHV provide mechanistic insights relevant to the PEC performance of GaN photoelectrodes with surface modifications. Nevertheless, under realistic PEC environments, the GaN (0001) surface could potentially undergo oxidation to form GaO or GaON layers, thereby leading to stronger upward band bending and modified interfacial charge-transfer dynamics. Future studies involving controlled surface oxidation (e.g., $O_2$ plasma treatment) and Pt/GaO(GaON)/GaN heterostructures will be valuable for elucidating the detailed interfacial charge-transfer dynamics and for bridging fundamental model investigations with practical PEC interfaces.

This work uncovers the ultrafast carrier relaxation dynamics in GaN-based photocatalysts and demonstrates how interfacial engineering via Pt modification fundamentally redefines the energetic landscape and transport behavior at the metal/semiconductor junction. By employing state-resolved tr-2PPE spectroscopy, we reveal that Pt/GaN enables ultrafast energy-independent charge injection from GaN to Pt within 50 fs and promotes bulk-to-surface electron transport by dynamically flattening the surface band bending via photoinduced surface photovoltage, effectively promoting charge separation and suppressing trap-state recombination. The interplay of these processes greatly improves charge separation and utilization, resulting in an approximately 6.6-fold enhancement in PEC-driven hydrogen evolution. By deepening our understanding of metal modification of semiconductor photoelectrodes, this work extends beyond conventional surface state suppression to include charge energy thermalization and transport dynamics at complex heterointerfaces. These insights offer a conceptual framework for the rational design of hybrid photocatalytic systems with tailored carrier pathways for efficient solar-to-chemical energy conversion.

## Methods

### Materials and reagents

Chemicals were used for PEC measurements: $Na_2SO_4$ (98%, Alfa Aesar), $K_3[Fe(CN)_6]$ (99.5%, Sinopharm Chemical). Deionized water (18.2 MΩ cm) was produced with a Millipore Q water purification system (Sigma-Aldrich). All chemicals were used as purchased without further purification. The used GaN (0001) films (thickness: 450 nm, size: $1 \times 1\,cm^2$) were purchased from HeFei Crystal Technical Material Co., Ltd., China.

### Sample preparation

Mg-doped GaN (0001) films with a thickness of 450 nm were epitaxially grown on sapphire substrates by hydride vapor phase epitaxy (HVPE) (HeFei Crystal Technical Material Co., Ltd, China). The GaN-coated wafers were sequentially cleaned by ultrasonic rinsing in acetone, ethanol, and deionized water for 10 minutes, followed by nitrogen drying. The samples were then mounted onto molybdenum holders and transferred into the preparation chamber of the 2PPE system under a base pressure of $-1 \times 10^{-10}$ mbar. Surface cleaning was performed by repeated cycles of $Ar^+$ sputtering (650 eV, 10 min) and annealing at 850 °C for 30 min in an ultrahigh vacuum (UHV) chamber, yielding an atomically ordered surface as confirmed by low-energy electron diffraction (LEED).

For surface Pt modification, the cleaned GaN wafers were transferred under vacuum to an electron-beam evaporation system (base pressure $-1 \times 10^{-9}$ mbar), where a 0.5 nm Pt film was deposited at a rate of $1\,Å\,s^{-1}$. The Pt film thickness was monitored using a quartz crystal microbalance (QCM). Post-deposition annealing at 500 °C for 1 h under UHV conditions was performed to enhance interfacial contact. The resulting surface morphology was characterized by atomic force microscopy (AFM).

### Time-resolved two-photon photoemission spectroscopy (tr-2PPE)

Ultrafast energy-resolved carrier dynamics were investigated using time-resolved two-photon photoemission spectroscopy (tr-2PPE) under ultrahigh vacuum (UHV) conditions at a base pressure of $-1.0 \times 10^{-10}$ mbar and room temperature (20.6 °C). A Coherent Vitara Ti:sapphire (Ti:Sa) oscillator coupled to a RegA 9050 amplifier (pumped by a Verdi 12 laser) provided pulses centered at 800 nm with a full width at half maximum (FWHM) of -50 fs and a pulse energy of

-10 μJ. The output beam was split, with part of the 800 nm light frequency-doubled to 400 nm in a β-barium borate (BBO) crystal, subsequently pumping two noncollinear optical parametric amplifiers (NOPAs). White-light continuum generated via sapphire disks seeded the NOPAs. One NOPA delivered visible (VIS) pulses tunable between 470–560 nm (2.64–2.21 eV), while the second NOPA output at 545 nm was frequency-doubled to generate ultraviolet (UV) pulses at 276 nm. Both VIS and UV pulses were compressed using a pair of quartz prisms to achieve -40 fs temporal resolution at the sample. The beams overlapped at the sample surface with a spot size of -100 × 100 μm², incident at an angle of -45° relative to the surface normal. Two neutral density filters were separately used to control the fluences of UV and VIS beams. Photoemitted electrons were collected by a time-of-flight (TOF) spectrometer positioned -3–5 mm from the surface, offering an angular acceptance of 7.3°, an energy resolution of -50 meV.

The UV beam was first split into two beams with an intensity ratio of 2:1 by a beam splitter before entering the UHV chamber. In the one-color configuration, the pump and probe beams possessed identical photon energies, corresponding to a super-bandgap excitation to investigate the subsequent hot carrier relaxation dynamics. At positive delay times, the weaker UV beam functioned as the pump and the stronger beam as the probe; at negative delay times, their roles were reversed. In the two-color configuration, the Vis beam served as the pump and the UV beam as the probe at positive delay times, with the roles reversed at negative delay times, enabling a sub-bandgap excitation that allowed selective probing of surface defect states at positive delay times. Auto-correlation (AC) and cross-correlation (CC) measurements of pump-probe were performed on a Cu (111) single crystal featuring an occupied surface state. The Cu surface was prepared by three cycles of Ar$^+$ sputtering (650 eV, $1.5 \times 10^{-5}$ mbar) followed by annealing at 527 °C for 20 min per cycle.

The transient dynamics of energy-resolved photoelectron intensity were analyzed by fitting to a model where the temporal response $S(t)$ is described as the convolution of a Gaussian instrument response function $G(t)$ accounting to pump-probe pulses with one or more exponential decay functions $E(t)$:

$$S(t) = G(t) \otimes E(t) \quad (1)$$

where $G(t) = \frac{1}{\sigma\sqrt{2\pi}} e^{-\frac{(t-t_0)^2}{2\sigma^2}}$ represents the Gaussian instrumental resolution function with $\sigma$ related to the instrument's temporal resolution, $E(t) = A^* e^{-\frac{t}{\tau}}$ is an exponential decay function characterized by time constant τ. For kinetic traces involving multiple relaxation processes for Pt/GaN sample at negative delay time, the total response is modeled as a sum of convoluted exponentials:

$$S(t) = \sum_i G(t) \otimes (A_i \times e^{-\frac{t}{\tau_i}}) \quad (2)$$

Here, $A_i$ and $\tau_i$ represents the amplitude and decay time of different processes, respectively. The Gaussian width $\sigma$ was independently determined from auto-correlation measurements for the one-color configuration and from cross-correlation measurements for the two-color configuration, respectively.

## X-ray photoelectron spectroscopy

X-ray photoelectron spectroscopy (XPS) was performed at room temperature (20 °C) using a SPECS Focus 500 monochromator coupled with a Phoibos 100 electron energy analyzer. An Al Kα X-ray source (photon energy 1486.74 eV) operated at 200–300 W was employed for excitation. Spectra were collected with a photoelectron take-off angle of 90° and a fixed angle of 54.7° between the X-ray source and the analyzer, using a pass energy of 10 eV. High-resolution core-level spectra were acquired with an energy step size of 0.05 eV and processed by applying Shirley background subtraction.

## Ultraviolet photoelectron spectroscopy

Ultraviolet photoelectron spectroscopy (UPS) was utilized to determine the surface work function and to probe the valence band structure of the samples. Measurements were performed at room temperature (20 °C) in a UHV chamber with a base pressure of approximately $10^{-10}$ mbar, using the same setup as for XPS characterization. Helium I (He-I) radiation (21.2 eV photon energy) generated from a discharge lamp operating at 25-35 W served as the ultraviolet excitation source. The incident beam impinged on the sample surface at an angle of 50° relative to the surface normal. To precisely identify the secondary electron cutoff, a bias voltage of −5.0 V was applied between the sample and the electron analyzer during the UPS measurements.

## Low-energy electron diffraction (LEED) measurements

Following in situ surface cleaning, the sample was transferred under UHV conditions to a separate chamber for low-energy electron diffraction (LEED) analysis. The LEED measurements were carried out at room temperature (20 °C) in a UHV environment with a base pressure better than $1 \times 10^{-10}$ mbar using a SPECS ErLEED 150 system. Incident electron energy of 226 eV was used to investigate the surface crystallographic structure. Diffraction patterns were recorded on a CCD detector and analyzed to assess surface periodicity and reconstruction.

## Surface photovoltage (SPV) measurements

Surface photovoltage (SPV) measurements were carried out using amplitude-modulated Kelvin Probe Force Microscopy (KPFM, Bruker Dimension Icon) under ambient conditions. A Pt/Ir-coated conductive AFM probe was employed to scan the sample surface. The lift height of the tip is 20 nm. The surface potential was recorded both in the dark and under illumination. A Xe lamp (300 W, PerfectLight Co., Ltd.) equipped with a monochromator was used as the illumination source. The SPV was obtained by calculating the difference in surface potential between the illuminated and dark states. This SPV signal reflects photoinduced changes in surface band bending and charge redistribution. To improve the signal-to-noise ratio, surface potential profiles were averaged over multiple scans. In our measurement system, a positive SPV indicates the accumulation of photogenerated holes at the surface, while a negative SPV corresponds to the transfer of photogenerated electrons to the surface.

## Transmission electron microscopy (TEM) and scanning transmission electron microscopy (STEM) measurements

Transmission electron microscopy (TEM) and scanning transmission electron microscopy (STEM) measurements were carried out using a Zeiss LIBRA 200 FE microscope operated at an accelerating voltage of 200 kV. The microscope was equipped with an omega-type energy filter, which was employed to acquire zero-loss filtered bright-field images, enhancing contrast and minimizing inelastic scattering contributions. A cross-sectional sample for TEM/STEM characterization was prepared using a Zeiss Crossbeam 340 focused ion beam (FIB) system with Ga ions, allowing precise site-specific thinning to produce electron-transparent sections for high-resolution imaging.

## Photoelectrochemical (PEC) measurements

The photoelectrochemical properties were evaluated using a CHI660E electrochemical workstation (CH Instruments) configured in a conventional three-electrode setup. The GaN photoelectrode served as the working electrode, with a platinum foil employed as the counter electrode and a saturated Ag/AgCl electrode used as the reference electrode. An aqueous solution of 0.5 M $Na_2SO_4$ (pH ≈ 6.8) was utilized as the electrolyte after thorough deaeration with high-purity nitrogen gas. The solution was stored in a sealed polypropylene container at room temperature (22 °C) in the dark to avoid contamination and degradation. Illumination was provided by a 300 W Xe arc lamp

equipped with a long-pass optical filter ($\lambda > 420$ nm) to simulate visible-light conditions. Linear sweep voltammetry (LSV) was performed to record the photocurrent density (J–V curves) over a potential range from $-0.8$ V to $0.8$ V versus the reversible hydrogen electrode (RHE) at a scan rate of $10$ mV s$^{-1}$. The measured potentials were converted to the RHE scale using the Nernst equation: $E_{RHE} = E_{Ag/AgCl} + 0.059 \times pH + 0.1976$. Photoelectrochemical impedance spectroscopy (PEIS) measurements (Biologic SP-300) were conducted at $-0.4$ V vs. RHE over a frequency range of $0.1$ Hz to $200$ kHz using a $5$ mV AC perturbation in $0.5$ M $Na_2SO_4$ solution. All PEC measurements were performed at room temperature ($25\,^\circ$C).

To evaluate the charge separation efficiency of the photoelectrode, we used $Fe^{3+}$ as a scavenger for the photogenerated electron due to fast kinetics and measured the photocurrent generated from $Fe^{3+}$ reduction ($J_{Fe^{3+}}$) in a $0.5$ M $Na_2SO_4$ electrolyte containing $5$ mM $K_3[Fe(CN)_6]$. The charge separation efficiency of the photoelectrode can be calculated using the following equation:

$$\eta_{separation} = \frac{J_{Fe^{3+}}}{J_{abs}} \qquad (3)$$

where $J_{abs}$ is the theoretical photocurrent density, which is taken to be $0.78$ mA/cm$^2$ for GaN film electrodes under $1.5$ AM illumination[19].

## Data availability
The data that support the findings of this study are provided in the Source Data file. Source data are provided with this paper.

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

## Acknowledgements

This research was supported by a grant from the National Program on Key Basic Research Project (No. 2021YFA1500600, F.F.), Fundamental Research Center of Artificial Photosynthesis (FReCAP, C.L., F.F.), the National Natural Science Foundation of China (22472170, Y.G., 22325205, F.F., and 22088102, F.F.), CAS Projects for Young Scientists in Basic Research (YSBR-004, F.F.), Fundamental Research Funds for the Central Universities (20720220011, F.F.), New Cornerstone Science Foundation through the XPLORER PRIZE (F.F.). Y.G. acknowledges the financial support from Dalian Institute of Chemical Physics, CAS, China.

## Author contributions

Y.G., F.F., and C.L. conceptualized the project. C.H. characterized XPS and UPS. Y.G. performed the experimental investigation, analyzed data, and wrote the manuscript. Y.X. carried out PEC experiments and analyzed the data. M.W. and H.K. performed TEM/STEM characterization. R.K. and D.F. supervised and developed the project. All authors discussed the results and commented on the manuscript.

## Funding

## Competing interests

The authors declare no competing interests.
