## [Transparent Peer Review file · Nature Communications]

Interface-driven energy-independent charge extraction in GaN photocatalysts

Corresponding Author: Dr Dennis Friedrich

Version 0:

Reviewer comments:

Reviewer #1

(Remarks to the Author)

This manuscript investigates ultrafast charge carrier dynamics at bare and Pt-decorated GaN surfaces using time-resolved two-photon photoemission spectroscopy (tr-2PPE). The authors show that Pt deposition effectively suppresses electron trapping at surface defects and enables energy-independent ultrafast charge transfer from GaN to Pt. In addition, Pt induces dynamic band flattening that enhances bulk-to-surface electron transport. These effects collectively improve charge separation and significantly enhance photoelectrochemical hydrogen evolution performance. The work is technically sound, well-structured, and experimentally thorough. It provides valuable insights into the role of co-catalysts in modulating carrier dynamics at semiconductor/metal interfaces. Overall, this study is a valuable and well-executed. With brief clarifications on surface condition and interface behavior under realistic environments, the manuscript will offer broader impact to the photoelectrochemical and photocatalysis communities.

1. The charge carrier dynamics studied under ultrahigh vacuum conditions may not fully reflect the interfacial behavior occurring in aqueous environments, as represented in the PEC experiments (e.g., Fig. 5). In particular, GaN (001) thin films grown on sapphire inevitably expose the polar c-plane, which is susceptible to oxidation in aqueous solution, forming GaO layers that can induce stronger upward band bending. It remains unclear whether similar charge transfer dynamics and band flattening occur in more realistic structures such as Pt/GaO/GaN or Pt/GaON/GaN. To better simulate the actual PEC environment, future studies could consider O₂ plasma treatment to form GaO or GaON termination instead of Ar sputtering, as it may more accurately replicate the surface chemistry of GaN in aqueous conditions. I suggest discussing the current limitations of this work and outlining future research directions for GaN/cocatalyst systems, particularly through probing carrier dynamics on more realistic surfaces.

2. The interfacial electron transfer from the GaN conduction band to Pt, which has a high work function, is somewhat expected. This raises an important question regarding the role of the cocatalyst's work function in facilitating charge extraction. Is high work function the key requirement for efficient electron transfer? As you noted in the introduction, a variety of co-catalysts (noble metals, transition metals, and metal oxides) have been employed on GaN PEC and photocatalytic systems. It would significantly enhance the depth of the manuscript if you could provide a more universal discussion on what properties of co-catalysts (e.g., work function, band alignment, chemical affinity, defect passivation ability) are critical from a charge carrier dynamics perspective.

Reviewer #2

(Remarks to the Author)

Here, the systematic regulation mechanism of Pt modification on the ultrafast electron dynamics of GaN surface was revealed, especially the discovery of an energy-independent electron injection process of ~50 fs and the revelation of its synergistic enhancement effect with bulk-surface transport. The topic is attractive to the broad readership of Nature Communications. So I recommend this manuscript can be accepted after revisions. And some questions should be addressed as following.

1. AFM shows that Pt exists in the form of nano-islands, but no coverage statistics or particle size distribution is provided. It is suggested to supplement the statistical information to determine whether the distribution of Pt is uniform and whether it forms continuous or discrete electron transmission channels.
2. Please add TEM/STEM imaging of the Pt/GaN interface.

3. Fig. 3g shows that the time constants range from 0.5 eV to 2.5 eV, which are indeed very close. However, the data points in the higher energy region (>2.5 eV) seem to be more scattered and show a slight upward trend. Please further explain whether this phenomenon is due to statistical errors or if there is indeed a slight energy dependence?
4. In the two-dimensional pseudo-color 2PPE (Fig 4b), the enhancement of low-energy signals of Pt/GaN during negative delay is attributed to "bulk phase transport". Whether this is affected by the inter-band transitions of Pt, considering that Pt has strong absorption in the UV-Vis region.
5. Lines 146-148: The low-energy surface defect states are correlated with Mg-doped defects. Line 169: The mid-bandgap defect states are directly attributed to nitrogen vacancies. However, there is a lack of direct characterization (such as EPR).
6. The photocurrent of Pt/GaN was increased by 6.6 times (Fig. 5c), but no stability data (such as the attenuation curve under continuous illumination) was provided. Further stability tests should be conducted. The Faraday efficiency should be supplemented to confirm the correspondence between H₂ production and current.
7. Is it possible to enhance the mechanistic interpretation through theoretical calculations. For example, predicting the density-of-states distribution at the Pt/GaN interface or modeling the influence of SPV on the band structure.

Reviewer #3

(Remarks to the Author)

Using the state-resolved tr-2PPE spectroscopy, the authors uncover the ultrafast carrier relaxation dynamics in GaN-based photocatalysts and demonstrate how interfacial engineering via Pt modification fundamentally redefines the energetic landscape and transport behavior at the metal/semiconductor junction. However, the manuscript needs further revisions to clarify the following points:

(1) What causes deep level defects at the Fermi level? Please provide a detailed explanation.

(2) Does the conclusion of 'energy independent charge extraction' hold true in other materials as well, or is it only observed in GaN?

(3) In the introduction section, the authors described that "Despite their impressive performance, the details of how such surface modifications alter the ultrafast energy relaxation pathways of photoexcited carriers are still unclear." The unclear question lies in which aspects, as research in this field is already very common. The author did not mention the significant significance and necessity of the research.

Version 1:

Reviewer comments:

Reviewer #1

(Remarks to the Author)

The authors have provided clear responses to my comments. The revised manuscript strengthens the logical flow of the discussion and acknowledges the limitations of UHV-based measurements when compared with realistic PEC environments. The added clarifications on possible surface oxidation and its influence on band bending and charge-transfer dynamics provide valuable context.

While additional experiments such as probing carrier dynamics on intentionally oxidized surfaces or comparing different cocatalysts with varied work function would deepen the mechanistic understanding, I believe that the current results already offer important scientific insights. The manuscript convincingly demonstrates key aspects of interfacial charge transport and the role of Pt in modulating carrier dynamics at GaN surfaces. Therefore, I recommend the manuscript for acceptance in its current form.

Reviewer #2

(Remarks to the Author)

I agree that it can be accepted for publication.

Reviewer #3

(Remarks to the Author)

The author has already answered the questions I raised.

REVIEWER COMMENTS

Reviewer #1 (Remarks to the Author):

This manuscript investigates ultrafast charge carrier dynamics at bare and Pt-decorated GaN surfaces using time-resolved two-photon photoemission spectroscopy (tr-2PPE). The authors show that Pt deposition effectively suppresses electron trapping at surface defects and enables energy-independent ultrafast charge transfer from GaN to Pt. In addition, Pt induces dynamic band flattening that enhances bulk-to-surface electron transport. These effects collectively improve charge separation and significantly enhance photoelectrochemical hydrogen evolution performance. The work is technically sound, well-structured, and experimentally thorough. It provides valuable insights into the role of co-catalysts in modulating carrier dynamics at semiconductor/metal interfaces. Overall, this study is a valuable and well-executed. With brief clarifications on surface condition and interface behavior under realistic environments, the manuscript will offer broader impact to the photoelectrochemical and photocatalysis communities.

1. The charge carrier dynamics studied under ultrahigh vacuum conditions may not fully reflect the interfacial behavior occurring in aqueous environments, as represented in the PEC experiments (e.g., Fig. 5). In particular, GaN (001) thin films grown on sapphire inevitably expose the polar c-plane, which is susceptible to oxidation in aqueous solution, forming GaO layers that can induce stronger upward band bending. It remains unclear whether similar charge transfer dynamics and band flattening occur in more realistic structures such as Pt/GaO/GaN or Pt/GaON/GaN. To better simulate the actual PEC environment, future studies could consider O₂ plasma treatment to form GaO or GaON termination instead of Ar sputtering, as it may more accurately replicate the surface chemistry of GaN in aqueous conditions. I suggest discussing the current limitations of this work and outlining future research directions for GaN/cocatalyst systems, particularly through probing carrier dynamics on more realistic surfaces.

2. The interfacial electron transfer from the GaN conduction band to Pt, which has a high work function, is somewhat expected. This raises an important question regarding the role of the cocatalyst's work function in facilitating charge extraction. Is high work function the key requirement for efficient electron transfer? As you noted in the introduction, a variety of co-catalysts (noble metals, transition metals, and metal oxides) have been employed on GaN PEC and photocatalytic systems. It would significantly enhance the depth of the manuscript if you could provide a more universal discussion on what properties of co-catalysts (e.g.,

work function, band alignment, chemical affinity, defect passivation ability) are critical from a charge carrier dynamics perspective.

Reviewer #2 (Remarks to the Author):

Here, the systematic regulation mechanism of Pt modification on the ultrafast electron dynamics of GaN surface was revealed, especially the discovery of an energy-independent electron injection process of ~ 50 fs and the revelation of its synergistic enhancement effect with bulk-surface transport. The topic is attractive to the broad readership of Nature Communications. So I recommend this manuscript can be accepted after revisions. And some questions should be addressed as following.

1. AFM shows that Pt exists in the form of nano-islands, but no coverage statistics or particle size distribution is provided. It is suggested to supplement the statistical information to determine whether the distribution of Pt is uniform and whether it forms continuous or discrete electron transmission channels.
2. Please add TEM/STEM imaging of the Pt/GaN interface.
3. Fig. 3g shows that the time constants range from 0.5 eV to 2.5 eV, which are indeed very close. However, the data points in the higher energy region (>2.5 eV) seem to be more scattered and show a slight upward trend. Please further explain whether this phenomenon is due to statistical errors or if there is indeed a slight energy dependence?
4. In the two-dimensional pseudo-color 2PPE (Fig 4b), the enhancement of low-energy signals of Pt/GaN during negative delay is attributed to "bulk phase transport". Whether this is affected by the inter-band transitions of Pt, considering that Pt has strong absorption in the UV-Vis region.
5. Lines 146-148: The low-energy surface defect states are correlated with Mg-doped defects. Line 169: The mid-bandgap defect states are directly attributed to nitrogen vacancies. However, there is a lack of direct characterization (such as EPR).
6. The photocurrent of Pt/GaN was increased by 6.6 times (Fig. 5c), but no stability data (such as the attenuation curve under continuous illumination) was provided. Further stability tests should be conducted. The Faraday efficiency should be supplemented to confirm the correspondence between H₂ production and current.
7. Is it possible to enhance the mechanistic interpretation through theoretical calculations. For example, predicting the density-of-states distribution at the Pt/GaN interface or modeling the influence of SPV on the band structure.

Reviewer #3 (Remarks to the Author):

Using the state-resolved tr-2PPE spectroscopy, the authors uncover the ultrafast carrier relaxation dynamics in GaN-based photocatalysts and demonstrate how interfacial engineering via Pt modification fundamentally redefines the energetic landscape and transport behavior at the metal/semiconductor junction. However, the manuscript needs further revisions to clarify the following points:

(1) What causes deep level defects at the Fermi level? Please provide a detailed explanation.

(2) Does the conclusion of 'energy independent charge extraction' hold true in other materials as well, or is it only observed in GaN?

(3) In the introduction section, the authors described that "Despite their impressive performance, the details of how such surface modifications alter the ultrafast energy relaxation pathways of photoexcited carriers are still unclear." The unclear question lies in which aspects, as research in this field is already very common. The author did not mention the significant significance and necessity of the research.

Responses to the reviewer(s)' comments

We have thoroughly revised our manuscript in accordance with the reviewers' comments, incorporated additional experimental evidence, and provide here a point-by-point response to all the comments.

Reviewer #1 (Remarks to the Author):

This manuscript investigates ultrafast charge carrier dynamics at bare and Pt-decorated GaN surfaces using time-resolved two-photon photoemission spectroscopy (tr-2PPE). The authors show that Pt deposition effectively suppresses electron trapping at surface defects and enables energy-independent ultrafast charge transfer from GaN to Pt. In addition, Pt induces dynamic band flattening that enhances bulk-to-surface electron transport. These effects collectively improve charge separation and significantly enhance photoelectrochemical hydrogen evolution performance. The work is technically sound, well-structured, and experimentally thorough. It provides valuable insights into the role of co-catalysts in modulating carrier dynamics at semiconductor/metal interfaces. Overall, this study is a valuable and well-executed. With brief clarifications on surface condition and interface behavior under realistic environments, the manuscript will offer broader impact to the photoelectrochemical and photocatalysis communities.

Response: We sincerely thank the reviewer for the positive and encouraging evaluation of our work. We fully agree that clarifying the surface condition and the interface behavior under realistic environments will further broaden the impact of this study. In the revised manuscript,

we have provided additional clarifications to strengthen the significance of our findings for the photoelectrochemical and photocatalysis communities, as detailed below.

1. The charge carrier dynamics studied under ultrahigh vacuum conditions may not fully reflect the interfacial behavior occurring in aqueous environments, as represented in the PEC experiments (e.g., Fig. 5). In particular, GaN (001) thin films grown on sapphire inevitably expose the polar c-plane, which is susceptible to oxidation in aqueous solution, forming GaO layers that can induce stronger upward band bending. It remains unclear whether similar charge transfer dynamics and band flattening occur in more realistic structures such as Pt/GaO/GaN or Pt/GaON/GaN. To better simulate the actual PEC environment, future studies could consider O₂ plasma treatment to form GaO or GaON termination instead of Ar sputtering, as it may more accurately replicate the surface chemistry of GaN in aqueous conditions. I suggest discussing the current limitations of this work and outlining future research directions for GaN/cocatalyst systems, particularly through probing carrier dynamics on more realistic surfaces.

Response: Thank reviewer for this insightful comment. We fully agree that photogenerated charge dynamics probed under ultrahigh vacuum (UHV) conditions may not completely replicate those in aqueous photoelectrochemical (PEC) environments, where GaN (0001) surfaces are prone to oxidation and may form GaO or GaON layers that modify the interfacial electronic structure. Such surface chemistry could indeed induce stronger upward band bending and alter the charge transfer dynamics compared with the UHV-prepared GaN surfaces studied here.

Nevertheless, we believe that our UHV-based measurements provide indispensable mechanistic insights at a well-defined Pt/GaN interface. Specifically, by detecting the kinetic

energy of photoemitted electrons in UHV, we are able to directly resolve the temporal- and energy-resolved carrier generation and transport processes. These measurements establish a mechanistic framework that is often inaccessible in aqueous PEC systems due to additional complexities arising from surface reactions and electrolyte interactions.

We also very much appreciate the reviewer's valuable suggestion regarding O₂ plasma treatment or other controlled oxidation strategies to deliberately form GaO/GaON terminations. We agree that such approaches could provide an effective means to better simulate the actual PEC surface chemistry of GaN electrodes, and we consider this as a promising direction for our future investigation.

In line with the reviewer's valuable feedback, we have added a dedicated discussion in the revised manuscript to explicitly acknowledge these limitations and to outline such future research opportunities, as below:

(P21) “The photogenerated charge separation and transport behaviors observed under UHV provide mechanistic insights relevant to the PEC performance of GaN photoelectrodes with surface modifications. Nevertheless, under realistic PEC environments, the GaN (0001) surface could potentially undergo oxidation to form GaO or GaON layers, thereby leading to stronger upward band bending and modified interfacial charge-transfer dynamics. Future studies involving controlled surface oxidation (e.g., O₂ plasma treatment) and Pt/GaO(GaON)/GaN heterostructures will be valuable for elucidating the detailed interfacial charge-transfer dynamics and for bridging fundamental model investigations with practical PEC interfaces.”

2. The interfacial electron transfer from the GaN conduction band to Pt, which has a high work function, is somewhat expected. This raises an important question regarding the role of the cocatalyst's work function in facilitating charge extraction. Is high work function the key requirement for efficient electron transfer? As you noted in the introduction, a variety of co-

catalysts (noble metals, transition metals, and metal oxides) have been employed on GaN PEC and photocatalytic systems. It would significantly enhance the depth of the manuscript if you could provide a more universal discussion on what properties of co-catalysts (e.g., work function, band alignment, chemical affinity, defect passivation ability) are critical from a charge carrier dynamics perspective.

Response: We fully agree that, while a high work function, as in the case of Pt, can thermodynamically facilitate electron transfer from the GaN conduction band, the efficiency of interfacial charge extraction is also determined by several additional factors. These include the relative band alignment between the semiconductor and cocatalyst, interfacial chemical bonding configuration, the ability of the cocatalyst to passivate surface states, surface morphology, and intrinsic catalytic activity toward interfacial reactions that play important roles in defining the rate and directionality of carrier transfer.

Considering these aspects, we have now expanded the discussion in the revised manuscript to provide a broader perspective on cocatalyst properties that influence electron extraction from GaN as below:

(P19) “Beyond the specific Pt/GaN system studied here, in which the metal and semiconductor exhibit distinct work functions, interfacial electron-transfer efficiency in GaN-based photocatalytic systems is largely dictated by a combination of cocatalyst properties. Specifically, the semiconductor/cocatalyst band alignment, the chemical nature of interfacial bonding, and the ability to passivate surface states collectively dictate the kinetics and selectivity of interfacial electron transfer⁴⁸. This broader perspective helps rationalize why a wide range of cocatalysts, including noble metals, transition metals, and metal oxides, can effectively facilitate carrier extraction on GaN surfaces.”

Reviewer #2 (Remarks to the Author):

Here, the systematic regulation mechanism of Pt modification on the ultrafast electron dynamics of GaN surface was revealed, especially the discovery of an energy-independent electron injection process of ~50 fs and the revelation of its synergistic enhancement effect with bulk-surface transport. The topic is attractive to the broad readership of Nature Communications. So I recommend this manuscript can be accepted after revisions. And some questions should be addressed as following.

Response: We are thankful to the reviewer for appreciating our work and constructive comments. All concerns raised have been carefully addressed, and the suggested revisions have been incorporated into the updated manuscript.

1. AFM shows that Pt exists in the form of nano-islands, but no coverage statistics or particle size distribution is provided. It is suggested to supplement the statistical information to determine whether the distribution of Pt is uniform and whether it forms continuous or discrete electron transmission channels.

Response: We thank the reviewer for this valuable suggestion. In response, we performed a statistical analysis of the Pt nano-islands based on AFM image over representative surface areas. The results (Figure R1) show that the Pt nano-islands have an average height size of ~0.73 nm with a surface coverage of ~56%. These data confirm that Pt predominantly remains in the form of discrete nano-islands rather than forming a continuous film. Moreover, the homogeneous distribution of nano-islands across the surface provides consistent interfacial contact, ensuring that charge transfer proceeds through localized metal/semiconductor junctions rather than continuous metallic channels.

These statistical results have been included in Supplementary Fig. 10a in the revised SI, and the corresponding discussion has been added to the revised manuscript as follows:

(P12) “Statistical analysis of AFM images (Supplementary Fig. 10a) reveals that Pt predominantly forms discrete nano-islands with an average height size of ~ 0.73 nm and a surface coverage of $\sim 56\%$. The homogeneous distribution across the surface indicates that charge transfer occurs primarily through localized metal/semiconductor junctions rather than continuous metallic channels.”

Figure R1. (a) AFM image of Pt nano-islands on GaN substrate. The colours denote different heights for Pt nano-islands and GaN terraces (purple). (b) Statistical analysis of height distributions of Pt nano-islands and their Gaussian fits.

2. Please add TEM/STEM imaging of the Pt/GaN interface.

Response: We have examined both TEM and STEM images of the Pt/GaN interface, as shown in Figure R2. The TEM results show that the Pt forms discrete nano-islands on the GaN surface, exhibiting intimate contact with the underlying GaN substrate. In addition, the STEM observations offer a more refined visualization of the interfacial region, highlighting a sharply defined, microscopically flat, and structurally uniform interface. The TEM/STEM results and

the corresponding discussion have been added to the revised manuscript to provide a more complete characterization of the Pt/GaN interface as follows:

(P12) “Transmission electron microscopy (TEM, Supplementary Fig. 10b,c) images show that the Pt nanoparticles are uniformly anchored on the GaN substrate with intimate interfacial contact. Moreover, scanning transmission electron microscopy (STEM, Supplementary Fig. 10d) image reveals a sharply defined, microscopically flat, and structurally uniform Pt/GaN interface.”

Figure R2. (a,b) Transmission electron microscopy (TEM) images of the Pt/GaN interface. (c) High-angle annular dark-field scanning transmission electron microscopy (HAADF-STEM) image of Pt/GaN interface.

3. Fig. 3g shows that the time constants range from 0.5 eV to 2.5 eV, which are indeed very close. However, the data points in the higher energy region (>2.5 eV) seem to be more scattered and show a slight upward trend. Please further explain whether this phenomenon is due to statistical errors or if there is indeed a slight energy dependence?

Response: Thank you for raising this insightful comment. In Fig. 3g, the time constants in the low- to mid-energy region (0.5–2.5 eV) are indeed very close, reflecting consistent carrier relaxation dynamics. For the higher energy region (>2.5 eV), the observed scatter and slight

upward trend are primarily attributed to larger statistical uncertainties due to weaker signal intensities and lower signal-to-noise ratios in this spectral window. Nevertheless, we note that this reflects a real energy-independent charge transfer, as the thermalization dynamics of hot electrons inherently exhibit significant energy dependence, regardless of whether they thermalize in Pt or in GaN. We have added a discussion in the revised manuscript to clarify this point:

(P13-P14) “Notably, the lifetimes of excited electron typically exhibit strong energy dependence in both semiconductors^{29,30,40} and metals^{41,42}. The absence of such dependence here provides compelling evidence for a uniform ultrafast energy-independent electron transfer from GaN to Pt, occurring within 41 fs across both DB and CB states.”

4. In the two-dimensional pseudo-color 2PPE (Fig 4b), the enhancement of low-energy signals of Pt/GaN during negative delay is attributed to “bulk phase transport”. Whether this is affected by the inter-band transitions of Pt, considering that Pt has strong absorption in the UV-Vis region.

Response: Thank you for this thoughtful comment. We agree that Pt exhibits strong interband absorption under the UV excitation used in our study (4.49 eV), which could in principle contribute to photoemission at negative delay. However, both Pt/GaN and bare GaN samples display nearly identical spectral shapes at negative delay times (Figure 4d), with the only distinction being an intensity enhancement in Pt/GaN. This strongly suggests that the signal characteristics are not governed by Pt-specific excitations. Moreover, kinetic analysis of the negative delay signal for Pt/GaN reveals a slower component with a lifetime of 1.96 ± 0.34 ps (Figure 4e), which is far longer than the typical tens-of-femtoseconds lifetime of hot carriers in Pt that rapidly decay via electron–electron scattering (*Phys. Rev. B*, **2001**, *63*, 115420). Taken

together, the consistent spectral features across both samples and the presence of a long-lived component provide compelling evidence that the observed signal does not originate from Pt interband transitions. Therefore, any contribution from Pt interband absorption can be considered negligible in our observed transient signals.

5. Lines 146-148: The low-energy surface defect states are correlated with Mg-doped defects.

Line 169: The mid-bandgap defect states are directly attributed to nitrogen vacancies. However, there is a lack of direct characterization (such as EPR).

Response: Following the reviewer's suggestion, we conducted additional EPR measurements to directly probe the defect states in GaN. As shown in Figure R3, a pronounced signal at a g value of 1.9798 is attributed to deep donor states associated with nitrogen vacancies, while a weaker signal at a g value of 2.0008 corresponds to shallow acceptor states induced by Mg doping (*Phys. Rev. B* **2002**, 65, 085312; *Mater. Sci. Eng. B*, **2002**, 93, 39–48). These EPR results provide direct experimental confirmation of the defect assignments proposed in our initial analysis. The corresponding data and the relevant discussion have been included in the revised manuscript as below:

(P8-P9) “In contrast, the transient feature at 0.5 eV is attributed to mid-gap DB states associated with nitrogen vacancies²⁸, as further supported by electron paramagnetic resonance (EPR) spectra revealing a pronounced deep donors signal related to nitrogen vacancies (Supplementary Fig. 8). These defect-related features in GaN (see Supplementary Fig. 9a for detailed defect distributions) differ significantly from the characteristic gradual population typically observed during defect-state trapping, as observed in Cu_2O ³¹.”

Figure R3. Electron paramagnetic resonance (EPR) spectra of GaN sample.

6. The photocurrent of Pt/GaN was increased by 6.6 times (Fig. 5c), but no stability data (such as the attenuation curve under continuous illumination) was provided. Further stability tests should be conducted. The Faraday efficiency should be supplemented to confirm the correspondence between H₂ production and current.

Response: We thank the reviewer for this constructive comment. In response, we conducted additional stability measurements on the Pt/GaN photoelectrode under continuous illumination. As shown in Figure R4, the photocurrent remains stable for over 800 min, demonstrating excellent operational durability of both samples. Furthermore, we evaluated the Faradaic efficiency for hydrogen evolution under the same conditions. The Pt/GaN photocathode exhibits a Faradaic efficiency exceeding 97% at -0.4 V vs. RHE, confirming that nearly all photogenerated electrons are utilized for the HER. These new results are presented in the revised SI (Figure S22b), and the corresponding discussion has been added to the revised manuscript as follows:

(P20-P21) “The Pt/GaN photocathode exhibits stable photocurrent for over 800 min under continuous illumination (Supplementary Fig. 22b), demonstrating excellent operational

durability. Moreover, the Faradaic efficiency for H₂ evolution exceeds 97% at -0.4 V vs. RHE, confirming that nearly all photogenerated electrons are consumed in the HER process.”

Figure R4. Stability test and Faradaic efficiency (solid circles) for GaN and Pt/GaN photocathodes for PEC hydrogen evolution reaction at the applied voltage of -0.4 V vs. RHE.

7. Is it possible to enhance the mechanistic interpretation through theoretical calculations. For example, predicting the density-of-states distribution at the Pt/GaN interface or modeling the influence of SPV on the band structure.

Response: We thank the reviewer for the suggestion. To provide further mechanistic support for charge transfer, we carried out simplified device-level simulations of the band bending and surface photovoltage (SPV) at the GaN surface under illumination. Our approach is based on solving the one-dimensional Poisson–drift–diffusion equations, which describe electrostatics and carrier transport in n-type GaN.

Specifically, the electrostatic potential ($\varphi(x)$) was obtained from Poisson’s equation:

$$\frac{d^2\varphi(x)}{dx^2} = -\frac{\rho(x)}{\varepsilon}$$

where $\rho(x) = q(p(x) - n(x) + N_D^+ - N_A^-)$ is the local space charge density, q is the elementary charge, and $\varepsilon = 9$ is the dielectric constant of GaN. N_D^+ and N_A^- denote the densities

of ionized donors and ionized acceptors, respectively. The electron and hole current densities are described by the drift–diffusion equations:

$$J_n(x) = q\mu_n n(x)E(x) + qD_n \frac{dn(x)}{dx},$$

$$J_p(x) = q\mu_p p(x)E(x) - qD_p \frac{dp(x)}{dx},$$

where $n(x)$ and $p(x)$ are the electron and hole concentrations, μ_n and μ_p are carrier mobilities, D_n and D_p are charge diffusion coefficients, and $E(x) = -d\phi(x)/dx$ is the electric field distribution. In this coupled framework, Poisson’s equation links the local potential to carrier densities, while the drift–diffusion and continuity equations determine how photogenerated carriers redistribute and screen the built-in field. Thus, these equations allow us to calculate both the equilibrium band bending (dark state) and the modified potential profile under illumination. The simulation results are presented in Figure R5. In the dark, the GaN surface shows an upward band bending of ~ 0.35 eV. Upon UV illumination, this band bending at the GaN surface is reduced by ~ 0.1 eV, in good agreement with our tr-2PPE observation of band flattening. Moreover, the calculated SPV increases with light intensity and gradually saturates at higher fluence, reflecting the effective screening of the built-in field by photogenerated carriers. We have added the corresponding figure in the SI (Supplementary Fig. 5d,e) and relevant discussion in the revised manuscript as below:

(P10) “This interpretation is further supported by simplified simulations (Supplementary Fig. 5d,e), which reproduce the experimentally observed band flattening and show that bandgap illumination reduces the upward band bending at the GaN surface by approximately 0.1 eV.”

Figure R5. (a) Band bending profiles at the GaN surface in the dark and under illumination, illustrating the band-flattening effect induced by photoexcitation. (b) Light-intensity-dependent SPV curve for the GaN surface under bandgap excitation, showing that the SPV increases with illumination intensity and gradually saturates.

Reviewer #3 (Remarks to the Author):

Using the state-resolved tr-2PPE spectroscopy, the authors uncovers the ultrafast carrier relaxation dynamics in GaN-based photocatalysts and demonstrates how interfacial engineering via Pt modification fundamentally redefines the energetic landscape and transport behavior at the metal/semiconductor junction. However, the manuscript needs further revisions to clarify the following points:

Response: We thank the reviewer for the positive assessment of our work. We have carefully revised the manuscript to clarify the points raised and believe that the changes improve both the clarity and impact of the study.

(1) What causes deep level defects at the Fermi level? Please provide a detailed explanation.

Response: We appreciate the reviewer for raising a point similar to that of *Reviewer #2* regarding the origin of deep-level defect states near the Fermi level. In n-type GaN, deep-level defect states near the Fermi level predominantly arise from intrinsic point defects and doping defects. In particular, nitrogen vacancies (V_N) are well known to act as deep donors, introducing electronic states near the conduction band minimum that are often pinned close to the Fermi level in moderately doped samples (*Appl. Phys. Lett.* **1996**, *69*, 2525–2527; *Appl. Phys. Lett.* **2012**, *100*, 142110; *Materials* **2024**, *17*, 1160). Additionally, Mg dopants can generate acceptor-like defect states that, when partially compensated, contribute to defect levels within the bandgap (*Phys. Rev. Lett.* **2015**, *114*, 016405). The coexistence of these donor- and acceptor-related defects gives rise to localized states near the Fermi energy. Importantly, we have performed EPR measurements that show a pronounced signal at $g = 1.9798$, consistent with nitrogen-vacancy-related deep donors, and a weaker signal at $g = 2.0008$ consistent with Mg-induced acceptors (Figure R3). The observed spectroscopic signatures indicate that the mid-gap features detected in tr-2PPE can be attributed to nitrogen-vacancy-related states. A detailed explanation has been incorporated into the revised manuscript as follows:

(P8-P9) “In contrast, the transient feature at 0.5 eV is attributed to mid-gap DB states associated with nitrogen vacancies²⁸, as further supported by electron paramagnetic resonance (EPR) spectra revealing a pronounced deep donors signal related to nitrogen vacancies (Supplementary Fig. 8). These defect-related features in GaN (see Supplementary Fig. 9a for detailed defect distributions) differ significantly from the characteristic gradual population typically observed during defect-state trapping, as observed in Cu_2O ³¹.”

(2) Does the conclusion of 'energy independent charge extraction' hold true in other materials as well, or is it only observed in GaN?

Response: We thank the reviewer for this important comment. The observation of energy-independent charge extraction has been experimentally demonstrated here for the Pt/GaN system. While the current results are specific to GaN, the underlying mechanism is likely applicable to other metal/semiconductor heterostructures with similar interface properties. Nevertheless, the universality of this phenomenon remains to be verified experimentally in other materials, as factors such as energy band structure, interfacial chemical bonding, and interface states may collectively influence the energy dependence of charge extraction. This work establishes a broadly applicable framework for elucidating ultrafast, energy-resolved interfacial charge dynamics in metal/semiconductor heterostructures, serving as a guiding paradigm for studies in alternative material platforms.

(3) In the introduction section, the authors described that “Despite their impressive performance, the details of how such surface modifications alter the ultrafast energy relaxation pathways of photoexcited carriers are still unclear.” The unclear question lies in which aspects, as research in this field is already very common. The author did not mention the significant significance and necessity of the research.

Response: We thank the reviewer for highlighting this point. While surface modification of semiconductors has been extensively explored, the ultrafast energy-resolved dynamics of photoexcited carriers at modified interfaces remain poorly understood. In particular, it is unclear how metal/semiconductor junctions alter the partitioning between nonthermal and thermalized electrons during charge transport, the timescales of energy relaxation, and state-specific charge separation mechanism. Our study addresses this gap by employing femtosecond time-resolved two-photon photoemission (tr-2PPE) spectroscopy, providing simultaneous temporal and energy-resolved insights into carrier dynamics. These measurements offer a direct mechanistic understanding of how interface engineering modulates nonequilibrium electron

behavior, thereby guiding rational design of cocatalysts and interfaces to maximize charge separation and photocatalytic efficiency. Thus, our study addresses an important gap in linking surface modification to ultrafast carrier behavior with energy and time resolution, which has not been comprehensively explored in previous reports.

We have revised the Introduction to better clarify the central research question and to highlight both the significance and necessity of this study as below:

(P3) “...most studies have focused on bulk carrier behaviors or averaged interfacial processes, whereas the femtosecond energy relaxation pathways at metal/semiconductor interfaces remain poorly understood. In particular, it is not well established how surface modifications alter the initial energy distribution, defect trapping, and interfacial injection of hot carriers immediately after photoexcitation. This knowledge gap is critical because only carriers that retain sufficient excess energy on ultrafast timescales can efficiently participate in catalytic reactions.”

(P4) “These hybrid catalysts benefit from engineered metal/semiconductor interfaces that optimize interactions between catalytic sites and reaction intermediates, reduce kinetic barriers, and improve charge separation. Despite these advances, the mechanistic details of how such metal/semiconductor interfaces reshape the ultrafast carrier energy landscape remain elusive. Specifically, the formation of a metal/semiconductor junction introduces new interfacial states^{14,22} that modulate the built-in electric field and band alignment^{23,24}, thereby influencing the carrier energy landscape and transport behavior. Yet these effects have rarely been studied with sufficient energy and time resolution sufficient to capture energy-resolved ultrafast charge dynamics.”

(P5) “Our results provide direct mechanistic insight into how interface engineering preserves hot carrier energy and enhances bulk-to-surface transport, offering a framework for rational design of cocatalysts and semiconductor interfaces to improve photocatalytic efficiency.”